# Pre-Conditioning Methods and Novel Approaches with Mesenchymal Stem Cells Therapy in Cardiovascular Disease

**DOI:** 10.3390/cells11101620

**Published:** 2022-05-12

**Authors:** Anthony Matta, Vanessa Nader, Marine Lebrin, Fabian Gross, Anne-Catherine Prats, Daniel Cussac, Michel Galinier, Jerome Roncalli

**Affiliations:** 1Department of Cardiology, Institute CARDIOMET, University Hospital of Toulouse, 31059 Toulouse, France; dr.anthonymatta@hotmail.com (A.M.); nader.e.vanessa@gmail.com (V.N.); lebrin.m@chu-toulouse.fr (M.L.); gross.f@chu-toulouse.fr (F.G.); galinier.m@chu-toulouse.fr (M.G.); 2Faculty of Medicine, Holy Spirit University of Kaslik, Kaslik 446, Lebanon; 3Department of Cardiology, Intercommunal Hospital Centre Castres-Mazamet, 81100 Castres, France; 4Faculty of Pharmacy, Lebanese University, Beirut 6573/14, Lebanon; 5CIC-Biotherapies, University Hospital of Toulouse, 31059 Toulouse, France; 6INSERM I2MC—UMR1297, 31432 Toulouse, France; anne-catherine.prats@inserm.fr (A.-C.P.); daniel.cussac@inserm.fr (D.C.)

**Keywords:** mesenchymal stem cells, preconditioning, exosome, engineered cardiac patches

## Abstract

Transplantation of mesenchymal stem cells (MSCs) in the setting of cardiovascular disease, such as heart failure, cardiomyopathy and ischemic heart disease, has been associated with good clinical outcomes in several trials. A reduction in left ventricular remodeling, myocardial fibrosis and scar size, an improvement in endothelial dysfunction and prolonged cardiomyocytes survival were reported. The regenerative capacity, in addition to the pro-angiogenic, anti-apoptotic and anti-inflammatory effects represent the main target properties of these cells. Herein, we review the different preconditioning methods of MSCs (hypoxia, chemical and pharmacological agents) and the novel approaches (genetically modified MSCs, MSC-derived exosomes and engineered cardiac patches) suggested to optimize the efficacy of MSC therapy.

## 1. Introduction

Several clinical trials have established the safety of mesenchymal stem cell (MSC) therapy and have shown promising results in the setting of cardiovascular disease over the past decades [1,2]. In ischemic heart disease, the role of existing conventional therapy, including percutaneous coronary intervention, coronary artery bypass graft and medical treatment, is limited to prevent future ischemic events and further expansion of myocardial damage [3]. Unlike MSC transplantation, there are no effects on myocardial repair, lost myocardial tissue and cardiomyocytes regeneration. Data from the literature showed a reduction in scar burden, myocardial fibrosis and infarct size, a reversion of left ventricular remodeling and an improvement in cardiac function after MSC therapy [1,4,5].

MSCs are undifferentiated, multipotent and self-renewable cells recognized for their potential of differentiation [6,7] and paracrine activity [2,8,9,10]. MSCs secrete diverse biological active cytokines, growth factors, chemokines and miRNA, resulting in anti-fibrotic, anti-inflammatory, regenerative, proliferative, immunomodulatory and angiogenic effects [11,12,13,14]. Neovascularization, angiogenesis, cardiomyocytes apoptosis inhibition, myocardial repair enhancement and dead cardiomyocytes replacement are the major targets of MSC therapy within the context of myocardial infarction [2]. MSCs are present in different human organs, but usually isolated from the following three main sources: umbilical cord, adipose tissue and bone marrow [2]. The latter is commonly used, despite the fact that it provides a mixture of non-purified miscellaneous cells [15]. After injection, MSCs are able to home, accumulate and engraft with the adjacent cellular components of the injured tissue and, subsequently, recruit additional progenitor cells [15,16]. However, hypoxia and increased free radical concentration in the context of myocardial infarction generate a detrimental microenvironment for transplanted MSCs [17]. Thus, preconditioning of MSCs with hypoxia or pharmacological or chemical agents in addition to novel strategies, such as exosome-mediated MSCs, genetically modified MSCs and engineered cardiac patches, were performed for improving the overall efficacy of MSC transplantation (Figure 1). All these techniques promote MSC survival and their capacity to form a regenerative and proliferative environment. Herein, we review the different preconditioning methods and novel approaches with MSCs in the setting of ischemic cardiac disease. 

## 2. Preconditioning Methods

### 2.1. Hypoxia-Preconditioned MSCs

The purpose of hypoxic preconditioning is to prolong the short survival time of grafted MSCs in the ischemic area, a major limitation of the therapeutic potential of stem cell therapy [18,19,20]. Indeed, hypoxic preconditioning increases the expression of protective factors against future hypoxic insult (hypoxia inducible factor-1 α (HIF-1α)), angiogenic factors (vascular epithelial growth factor, angiopoietin-1 and erythropoietin), pro-survival proteins (P65, P50 and P105) and anti-apoptotic proteins (Bcl-xl et Bcl-2) [21]. Previous study results showed that 24 h hypoxia exposure could dramatically amplify MSC proliferation and reduce their apoptosis by mainly activating the HIF-1α/Apelin/APJ axis [22]. First, HIF-1α modulates oxygen homeostasis and promotes cell function and tolerance in a hypoxic microenvironment [23]. It plays a crucial role in cardiomyocytes protection against ischemia-reperfusion injury by regulating mitochondrial reactive oxygen species [24] and heme oxgenase-1 [25]. Then, the inhibition of inflammatory reaction and apoptosis, up-regulation of collagen matrix and glycolysis, stimulation of angiogenesis and improvement of oxygen delivery are mediated by HIF-1α [26]. Second, the stimulation of Apelin/APJ enhances MSC survival and differentiation [27]. Moreover, hypoxia preconditioning activates other pathways, such as SDF-1α/CXCR4 axis implicated in MSC migration, detention and homing [28,29], PI3K/Akt signaling pathway that blocks cell death [30] and GRP78 that interferes in angiogenic cytokine secretion and cell migration [31]. A recent study revealed that extracellular vesicles from hypoxia-preconditioned MSCs may partly alleviate myocardial injury by targeting the thioredoxin-interacting protein-mediated HIF-1α pathway [32]. The evidence suggests that transplantation of hypoxia-preconditioned MSCs in the setting of myocardial infarction results in better cardiovascular outcomes by enhancing MSC engraftment, proliferation, differentiation, survival and paracrine activity [33,34]. Furthermore, it has shown that hypoxic preconditioning enhances survival and proangiogenic capacity of human first trimester chorionic villus-derived MSCs for fetal tissue engineering [35]. Lastly, we spotlight that different percentages of hypoxia have different outcomes. For example, 1% hypoxia extends MSC lifespan and maintains their proliferation rate [36,37]. In addition, 2% and 5 % hypoxia increased MSC number and viability [34]. Upregulation of stemness-related genes was observed with 3% hypoxia [38,39]. In other words, severe hypoxia (<1%) activates glycolytic metabolism and induces MSC quiescence, whereas moderate hypoxia (3–5%) stimulates MSC proliferation [40,41,42]. Although, short duration exposure to hypoxia (24 h) yields a better result than that of longer duration (72 h). 

### 2.2. Preconditioning with Pharmacological and Chemical Agents

Numerous growth factors, drugs and pharmacological and chemical substances have been used for MSC preconditioning (Table 1). For example, the treatment of MSCs with IGF-1 showed a positive impact on survival, detrimental infarct consequences (infarct size, ventricular remodeling and fibrosis) and pro-inflammatory cytokines [43]. HGF promotes MSC differentiation into cardiomyocytes, whereas the effect of IGF-1 on MSC potential of differentiation remains uncertain [44,45]. On the other hand, pretreatment with bFGF improves stem cells’ homing ability to the infarct zone and angiogenesis [46]. The pretreatment of MSCs with growth factor combinations (FGF-2, IGF-1 and BMP-2) leads to stronger engraftment, better viability in hypoxic situations, enhanced cell to cell communication and greater cytoprotective effects [47]. The results of a recent study showed superior cardiac function recovery and vasculogenesis in the infarcted myocardium 6 weeks after an injection of treated MSCs with SDF-1α in a rat model [48]. Beyond growth factors, variant biological active substances have been tested to improve the therapeutic efficacy of MSC therapy. Indeed, MSC pretreatment with angiotensin II potentiates the paracrine activity, angiogenesis, gap junction formation and global clinical outcome, by up-regulating the expression of VEGF, Cx43 with no effects on the differentiation mechanisms [49]. In addition, the left ventricular cardiac function and cardiomyogenic transdifferentiation have been significantly improved after transplantation of pioglitazone pretreated MSCs [50]. Thus, it seems a promising preconditioning method to predict cardiomyogenesis. Furthermore, pretreatment of MSCs with atorvastatin significantly improved cardiac function, reduced infarct size, decreased serum marker level of inflammation and fibrosis, inhibited apoptosis and enhanced survival of implanted MSCs, via activating the subtype eNOS of nitric oxide synthase [51]. Atorvastatin also improved the migration capacity of MSCs by increasing the expression of CXCR4 [52]. Benefits on MSC survival and differentiation have been observed with simvastatin pretreated MSCs [53]. Statin pretreatment positive outcomes have been also observed after transplantation of sevoflurane-preconditioned MSCs, which increase the expression of HIF-1α, HIF-2α, VEGF and p/Akt/Akt [54]. Transplantation of LPS- (lipopolysaccharide) preconditioned MSCs in the setting of myocardial infarction improves their biological and functional characteristics by up-regulating VEGF, phosphorylated Akt and TLR4 pathway [55]. Thereby, longer survival of transplanted cells, intense neovascularization and greater amelioration of left ventricular ejection fraction have been reported [55]. Vitamine E decreases oxidative stress and H_2_O_2_-related senescence by up-regulating the expression of VEGF, TGF-β and LDH [56]. The proliferation ability of MSCs has been promoted with astragaloside IV by inhibiting the translocation of NF-*k*Bp65 [57], apple ethanol extract by inducing the phosphorylation of eIF4E, p44, p70S6K, MAPK, eIF48, p44/42, mTOR and S6RP [58], oxytocin by activating the Akt/ERK1/2 axis [59], LL-37 by activating the MAPK pathway [60] and migration inhibitory factor by releasing VEGF, BFGF, HGF and IGF [61]. Although, the migration and homing abilities of MSCs have been improved with deferoxamine by expressing HIF-1α, CXCR4, CCR2, MMP-2 and MMP-9 [62], IL-1β by producing different cytokines, chemokines and adhesions molecules [63] and TGF-β1 by triggering the canonical SMADs [64]. In addition, the improvement of cardiovascular stem cell therapeutic outcomes has been associated with transplantation of 2,4-dinitrophenol [65], oxytocin [66] and dimethyloxalyglycine [67] pretreated MSCs. Finally, our group has shown that melatonin (pineal hormone to protect tissue from oxidative damage) pretreated MSCs modulate survival, differentiation and antifibrotic activity of cardiac fibroblasts [68]. Our results showed that MSCs significantly improved morphological and functional cardiac parameters two weeks after injection. However, the partial recovery of ventricular ejection fraction was maintained up to two months only when MSC survival was increased by melatonin treatment. These data indicate that the increased number of viable cells is critical for the amplification of the beneficial effects of MSCs on injured myocardium and ventricular function recovery. These properties of MSCs opened new perspective for understanding the mechanisms of action of MSCs and anticipated their potential therapeutic effects.

## 3. Novel Approaches

### 3.1. Genetic Modification of MSCs

Genetic modification of MSCs up-regulates the expression of specific genes implicated in MSC migration, adhesion, survival and premature senescence (Table 2). To begin, the migratory ability of MSCs has been promoted by overexpressing nuclear receptors (Nur1, Nur77) [69,70], integrin subunit-α4 [71], aquaporin-1 [72] and CXCR4/CXCR7 that serve as receptors for major cellular migratory process chemokine (SDF-1) [73,74]. Then, the overexpression of α(1,3)fucosyltransferase [75], focal adhesion kinase [76], integrin-linked kinase [77] and miR-9-5p [78] have been linked to stronger MSC adhesion and engraftment. However prolonged survival of transplanted MSCs has been demonstrated with overexpression of integrin-linked kinase that activates AKT, mTOR, JAK2/STAT3 signaling pathways [79,80], protein kinase Cε [81], Trkβ [82] and Gremlin1 [83]. The up-regulation of Sox2 and Oct4 genes accelerates cell transition from phase G1 into phase S, enhancing MSC proliferation, differentiation and anti-inflammatory effect [84,85]. EphB2 overexpression reduced premature senescence by suppressing mitochondrial reactive oxygen species accumulation, which triggers MSC senescence [86]. Transplantation of Kallikrein-1 genetically modified MSCs attenuates cardiac inflammation, cardiomyocytes apoptosis and myocardial fibrosis via VEGF, GSK-3β and NO signaling pathways activation [87,88,89,90,91]. Thus, pleiotropic, angiogenic proteolytic and cardioprotective effects have been attributed to Kallikrein-1 [91]. In the context of acute myocardial infarction, several clinical trials have demonstrated the therapeutic benefits of transplantation of genetically modified MSCs in animal models. For example, the target outcomes of MSC therapy were maintained for longer durations with transplanted Akt or angiopoietin1-MSCs [92]. Although, an injection of Bcl-2 or SDF-1α-or TNFR gene modified MSCs or miR-377 depleted MSCs potentiates the required efficacy of vascular density, cardiac function, infarct size and myocardial fibrosis [93,94,95,96,97,98]. Genetic modification of MSCs is applied using viral vectors, such as adenoviral, lentiviral and retroviral vectors for nucleic acid delivery [99], non-viral delivery systems, such as plasmid DNA, polymers, nanoplasmids, liposomes and DNA minicircles [100,101,102] and the novel gene-editing technology, clustered regularly interspaced short palindromic repeats (CRISPR/Cas9) [103]. This last technique allows one to insert a new sequence in the genome via homology-directed repair, which could rectify an acquired gene mutation or provoke a knock-in or knock-out mutation or suppress a specific gene expression [103].

### 3.2. MSCs Derived-Exosomes

Exosomes are classified as extracellular vesicles that are continuously produced and released by various hematopoietic and non-hematopoietic cells [104,105,106]. Exosomes interfere in variant cell to cell interaction pathways that are implicated in different physiological and pathological patterns [107]. Endocytosis, membrane fusion and membrane receptors represent the three exosomal mechanisms to regulate cell to cell communication [108]. Exosomes are mainly isolated for therapeutic application, either by ultrafiltration or ultracentrifugation-based methods [107]. Preclinical experimental animal models have demonstrated the therapeutic benefits of MSC-derived exosomes in the setting of myocardial infarction. An injection of MSC- derived miRNA-enriched exosomes have showed remarkable outcomes, such as reduction in infarct size and myocardial fibrosis with miR-22 via acting on MECP2 [109], enhancement of anti-apoptotic and cardioprotective effects with miR-221 by inhibiting PUMA expression [110], promotion of cardiac function recovery with miR-19a by suppressing PTEN and activating ERK pathways, respectively [111], and improvement in angiogenesis with miR-210 [112]. Overall, the transplantation of exosomes-derived MSCs leads to stronger cardioprotective effects [113] and reduction in the risk of tumorigenicity [114] than MSC-based therapies.

### 3.3. Engineered Cardiac Patches

Cell sheets and cell containing scaffolds represent the two forms of engineered cardiac patches [115]. Multiple cell types, such as endothelial cells, cardiac fibroblasts, pluripotent stem cells, cardiomyocytes, progenitor cells and smooth muscle cells have been incorporated into engineered cardiac patches [116,117,118,119]. Consequently, the replacement of damaged cardiomyocytes with functional cardiac cells is the ultimate target of engineered cardiac tissue transplantation. Promising results with evidence of remuscularization of the fibrotic myocardium have been demonstrated in numerous pre-clinical studies [120,121,122,123,124,125]. Indeed, cardiac function recovery has been observed in rats and minipigs after 4 weeks of transplantation of cell-free patches in the setting of acute anterior myocardial infarction [126]. Furthermore, the implantation of a bioengineered cardiac patch has shown superior therapeutic efficacy compared to that of decellularized placenta and human-induced pluripotent stem cells for myocardial repair, mediated by growth and pro-angiogenic factors that promote engraftment, neovascularization and paracrine function [127]. However, the need for a huge quantity of exogenous cardiac cells to refill the injured myocardium and stable electromechanical coupling between the transplanted cardiac patches and host tissue for long-term engraftment are the main challenges for this novel cardiac approach [128]. Thereby, larger and thicker vascularized cardiac patches that are synchronized with the circulatory and electromechanical systems of the native myocardium are required to overcome these limitations. The safety concern of cardiac patch therapy was limited to arrhythmias, which were generally transient and non-fatal [129,130,131]. It is noteworthy that a recently published study has revealed the efficacy of upscaled engineered heart tissue to improve left ventricular function and reduce the infarct size in the context of ischemic myocardial disease without documenting a significant difference in arrhythmogenicity, compared to a cell-free patch group in a rabbit model [132].

Overall, the therapeutic benefits of MSCs have been demonstrated in the treatment of ischemic cardiomyopathies [133]; however, the limited engraftment and poor survival of MSCs injected into an ischemic heart hindered the efficacy of the treatment. The use of scaffolds and polymeric supports to provide transplanted cells anchorage, a straightforward approach to circumvent this limitation, has already been tested [134]. Indeed, a robust therapeutic benefit of ADSCs when transplanted with a collagen scaffold in a preclinical porcine model of myocardial infarction, compared with cells without a collagen scaffold, has been successfully demonstrated. The functional improvement in cardiac function and myocardial remodeling after ADSC-collagen scaffold transplantation was associated with increased cell engraftment [135]. The positive preclinical results obtained using different biomaterials and cell types invited researchers to test whether these experimental procedures could be translated into the clinical setting. Thus, the phase I MAGNUM clinical trial was designed with the purpose of comparing the effects exerted by bone-marrow mononucleated cells-seeded cellularized collagen matrices with those exerted by cells alone, in patients presenting left ventricular post-ischemic myocardial scars. The results were promising because no treatment-related serious adverse events were reported during the follow-up period and heart functionality and mechanical parameters improved significantly in patients who received the cellularized patches. In other words, clinically, this procedure seems to be safe, feasible, and effective [136]. We mention that one of the first clinical trials on engineered heart muscle in patients with terminal heart failure is ongoing, BioVAT-HF (ClinicalTrials.gov: NCT04396899). However, a recent report of in-human transplantation of an allogenic-induced pluripotent stem cell-derived cardiomyocytes patch into the epicardium of the anterior and lateral walls via the fourth intercostal space in a patient with ischemic cardiomyopathy has been currently published [137]. This report signals the safety and efficacy of these patches on NYHA class, left ventricular end systolic volume and Vo2 peak at the 1-year follow-up after transplantation [137]. Moreover, the ESCORT trial on six patients referred to cardiac surgery has also demonstrated the technical feasibility of producing clinical-grade human embryonic stem cell-derived cardiovascular progenitors delivered in a fibrin epicardial patch, and supported their short- and medium-term safety, thereby, setting the grounds for adequately powered efficacy studies [138]. Finally, the translation of preclinical findings to the first clinical results requires the creation of cardiac scaffolds following all the GMP regulatory and quality requirements in order to test their safety as potential therapeutic products. The CARDIOPATCH Interreg Sudoe program aims to create a 2.0 version patch (v2.0) with growth factors and genetically improved mesenchymal cells and iPS-derived cardiac cells that improve cell survival of both the implanted cells and the ischemic cardiac tissue, as well as their pro-angiogenic capacity.

## 4. MSCs Perspectives

As is known for most new therapies, the progression of MSC therapy has been hard, slow and punctuated by difficulties. The available evidence proves the safety of MSC transplantation, which represents a new, hopeful strategy for the management of cardiovascular disease, particularly ischemic and non-ischemic heart failure [139,140,141]. Up to date, numerous Phase I and Phase II trials have demonstrated promising results with regenerative medicine in the setting of heart failure and myocardial infarction [2]. The findings from these trials are divergent. However, several important points have not yet been defined, such as the preferred cell source, preparation method, appropriate dose and recommended manner of administration. Defining these parameters constitutes an important step towards establishing a standard approach with MSC therapy and ensuring result reproducibility. The results from pivotal phase III trials are required to support the clinical application of MSC therapy in the cardiovascular field. Recently, stem cell therapy was approved for the management of complex perianal fistulas in Crohn’s disease [142]. We emphasize that pre-conditioning methods have contributed to overcome numerous hurdles, such as injected cell migration, engraftment, proliferation, differentiation and survival, resulting in stronger efficacy and better outcomes. Furthermore, recent studies have proved the benefits of mechanical stimulation on MSCs and the surrounding microenvironment and showed the interest of its application for bone regeneration therapy [143]. Lastly, engineered cardiac patch technology represents a revolution in stem cell therapy for cardiovascular disease, but manufacturing larger and thicker constructs that are suitably vascularized and incorporated with the electromechanical and circulatory systems of the vernacular myocardium is necessary for the clinical translation step.

## 5. Conclusions

To conclude, transplantation of pre-conditioned MSCs results in better therapeutic efficacy in the setting of cardiovascular disease, especially with moderate hypoxia pre-conditioning. In parallel, the available novel techniques are able to overcome the limitations (MSCs homing ability, engraftment and survival) of this regenerative medicine, promoting stronger cardiovascular outcomes. Starting translational engineered cardiac patch practice from pre-clinical trials in animal models to in-human trials may change our future management of heart failure.

## Figures and Tables

**Figure 1 cells-11-01620-f001:**
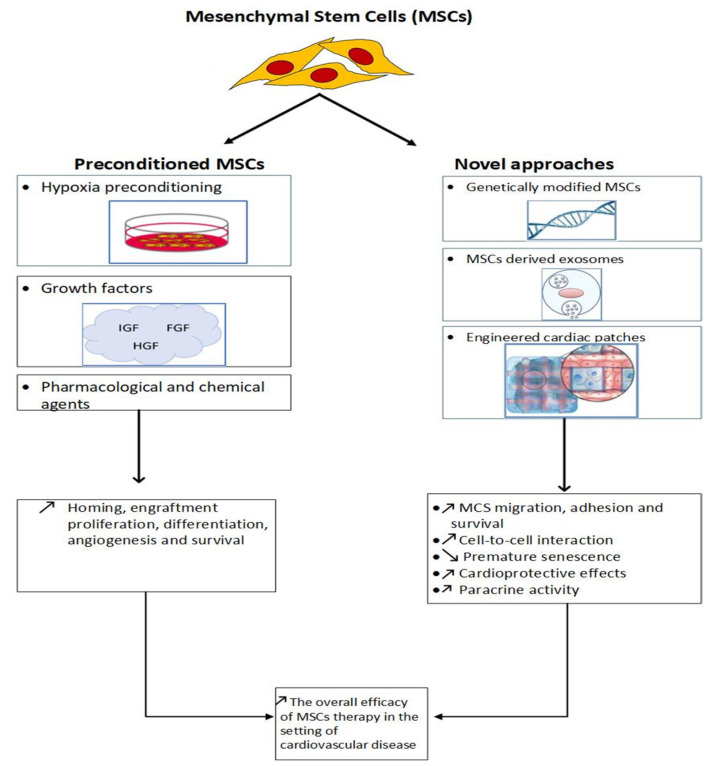
Figure illustrating mesenchymal stem cell (MSC) preconditioning methods, novel approaches and their main impacts.

**Table 1 cells-11-01620-t001:** MSC preconditioning with pharmacological and chemical agents.

Agents	Effects on	References
IGF-1	survival, infarct consequences, pro-inflammatory cytokines	[43]
HGF	differentiation into cardiomyocytes	[44,45]
bFGF	stem cells homing and angiogenesis	[46]
FGF-2, IGF-1 and BMP-2 combination	engraftment, viability, cell to cell communication, cytoprotective effect	[47]
SDF-1α	cardiac function recovery and vasculogenesis	[48]
Angiotensin II	paracrine activity, angiogenesis and gap junction formation	[49]
Pioglitazone	cardiac function and cardiomyogenic trans differentiation	[50]
Atorvastatin	cardiac function, infarct size, serum markers level of inflammation and fibrosis, apoptosis, migration capacity and survival of implanted MSCs	[51,52]
Simvastatin	MSC survival and differentiation	[53]
Sevoflurane	homing, survival and differentiation	[54]
LPS (lipopolysaccharide)	biological and functional characteristics of MSCs	[55]
Vitamine E	decreases oxidative stress and H2O2-related senescence	[56]
Astragaloside	proliferation ability of MSCs	[57]
Apple ethanol	proliferation ability of MSCs	[58]
Oxytocin	proliferation ability of MSCs	[59]
LL-37	proliferation ability of MSCs	[60]
Deferoxamine	migration and homing abilities of MSC	[62]
IL-1β	migration and homing abilities of MSCs	[63]
TGF-β1	migration and homing abilities of MSCs	[64]
2,4-dinitrophenol	cardiovascular stem cell therapeutic outcomes	[65]
Oxytocin	cardiovascular stem cell therapeutic outcomes	[66]
Dimethyloxalyglycine	cardiovascular stem cell therapeutic outcomes	[67]
Melatonin	survival, differentiation and antifibrotic activity	[68]

**Table 2 cells-11-01620-t002:** Outcomes of genetic modifications of MSCs.

Function	Up-Regulating Genes	References
Improved MSC migration	Nur1, Nur7	[69,70]
	Integrin subunit- α4	[71]
	Aquaporin-1	[72]
	CXCR4/VXCR7	[73,74]
Improved MSC adhesion and engraftment	α(1,3)fucosyltransferase	[75]
	Focal adhesin kinase	76]
	Integrin-linked kinase	[77]
	miR-9-5-p	[78]
Prolonged MSC survival	Integrin-linked kinase	[79,80]
	Protein kinase Cε	[81]
	Trkβ	[82]
	Gremlin 1	[83]
Enhanced MSC proliferation and differentiation	Sox2 and Oct4	[84,85]
Reduced premature senescence	EphB2	[86]
Sustained therapeutic efficacy	AktAngiopoietin 1	[92]
Better outcomes in setting of acute myocardial infarction	Bcl-2	[93]
	SDF-1α	[95]
	TNFR	[97]
	miR-377	[98]

## Data Availability

Not applicable.

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
