# Peer review of "Pre-Conditioning Methods and Novel Approaches with Mesenchymal Stem Cells Therapy in Cardiovascular Disease"

_cells, 2022, doi:10.3390/cells11101620_

Round 1

Reviewer 1 Report

Several sentences along the manuscript are not clear (99-101, 163-164, among several others)

Some errors (futur, angoiogenesis, peconditionig)

It is difficult to follow the ideas. It is a descriptive article of many original articles missing connection between sentences/facts. It is not an instructive/for reflection article. 

It reviewed an excessive number of articles. Each sentence has its reference, I could just appreciate the author's explanation in lines 119-129.

The topic (preconditioning methods) is well described in the literature. 
The pre-conditioning of MSCs with hypoxia is a huge world that was not well resumed in section II. Different percentages of hypoxia have different outcomes: severe hypoxia induces quiescence of MSCs, moderate hypoxia stimulates proliferation of MSCs. 

The article is not just focused on approaches with MSCs as the title announce, as the first paragraph of engineered cardiac patches.

A review article could be improved with figures. 

Author Response

Reviewer #1:

Point 1: Several sentences along the manuscript are not clear (99- 101, 163-164, among several others):

Response 1: Thank you for your positive evaluation and for your suggestions. Sentences in line 99-101 and 163-164 were reformulated.

Point 2: Some errors (futur, angoiogenesis, peconditionig):

Response 2: Abstract and whole manuscript were reviewed by native English speaker for typos, grammatical errors and potential improvement.

Point 3: It is difficult to follow the ideas. It is a descriptive article of many original articles missing connection between sentences/facts. It is not an instructive/for reflection article.

It reviewed an excessive number of articles. Each sentence has its reference, I could just appreciate the author's explanation in lines 119-129:

Response 3: We have tried to review the maximum possible of relevant published articles in literature in order to present a valuable review highlighting the main available data on this topic. We acknowledge an excessive number of references that justify the purpose of the review highlighting some novelties and added a perspective section. We also added the figure to summarize the main points for the readers and 2 tables for sub-sections.

Point 4: The topic (preconditioning methods) is well described in the literature.
The pre-conditioning of MSCs with hypoxia is a huge world that was not well resumed in section II. Different percentages of hypoxia have different outcomes: severe hypoxia induces quiescence of MSCs, moderate hypoxia stimulates proliferation of MSCs:

Response 4: The paragraph concerning hypoxic pre-conditioning was changed according to your recommendation and developed taking into consideration the impact of different percentages of hypoxia.

Point 5: The article is not just focused on approaches with MSCs as the title announce, as the first paragraph of engineered cardiac patches:

Response 5: The title was modified to involve the preconditioning methods and novel approaches which were both treated in the text.

Point 6: A review article could be improved with figures:

Response 6: A figure was added. 

Reviewer 2 Report

The authors provide a mini-review on the potentiation strategies for enhancing MSC therapeutic efficacy in cardiovascular disease. The contents of the review fit well the topic of the Special issue and is of interest for the readership of this journal. However, some similar review articles summarized recent progress on this topic (e.g. Stem Cell Rev Rep, 2013, 9:281-302 ; Mol Ther, 2018, 26:1610-1623), which appears to limit somewhat the novelty of this review. So it is strongly suggested to highlight the novelty of this paper and focus on the novel approaches for enhancing MSC therapeutic efficacy in cardiovascular disease. The review should provide in-depth viewpoints, and would benefit from a more extensive paragraph on perspectives, elaborating in more detail the current state as well as the future challenges. Moreover, more and more studies have proved that physical factors (e.g. mechanical stimulation) could enhance MSC therapeutic efficacy, these progress also should be provided.

In addition, the authors mentioned figure on Page 2, Line 50, however, this reviewer do not find any figures in this article. In order to enhance the understanding of the review, it is suggested to include additional display items, e.g. one table summarizing preconditioning methods, another table summarizing novel approaches, one figure illustrating the potentiation strategies for MSCs in cardiovascular disease.

Furthermore, I'm also worried about the novelty of the information in this manuscript– only 16 (13%) out of 128 total references from the last three years (2019-2021), which does not fit well with the dynamics in the field.

Author Response

Reviewer #2:

Point 1: The authors provide a mini-review on the potentiation strategies for enhancing MSC therapeutic efficacy in cardiovascular disease. The contents of the review fit well the topic of the Special issue and is of interest for the readership of this journal.

Response 1: We thank the reviewer for his positive feedback and his support for the interest of this review to the concerned special issue.

Point 2: However, some similar review articles summarized recent progress on this topic (e.g. Stem Cell Rev Rep, 2013, 9:281-302 ; Mol Ther, 2018, 26:1610-1623), which appears to limit somewhat the novelty of this review. So it is strongly suggested to highlight the novelty of this paper and focus on the novel approaches for enhancing MSC therapeutic efficacy in cardiovascular disease. The review should provide in- depth viewpoints, and would benefit from a more extensive paragraph on perspectives, elaborating in more detail the current state as well as the future challenges.

Response 2: Thank you for your positive feedback and for the effort that you have dedicated to review this manuscript. You have raised the important points of the review. As recommended by the reviewer, a more extensive paragraph on perspectives was added in order to highlight current state and future approaches.

Point 3: Moreover, more and more studies have proved that physical factors (e.g. mechanical stimulation) could enhance MSC therapeutic efficacy, these progress also should be provided.

Response 3: Physical factors like mechanical stimulation are more investigated in bone disease than cardiovascular disease, but we agree with you that it also represents a novelty of MSCs therapy. A related sentence has been added.

Point 4: In addition, the authors mentioned figure on Page 2, Line 50, however, this reviewer do not find any figures in this article. In order to enhance the understanding of the review, it is suggested to include additional display items, e.g. one table summarizing preconditioning methods, another table summarizing novel approaches, one figure illustrating the potentiation strategies for MSCs in cardiovascular disease.

Response 4: We submit the figure illustrating the pre-conditioning methods and novel approaches with the main outcomes. Thus, we follow your suggestion by adding an illustrative figure of the main findings resuming the major points of this review. We were also able to add two tables summarizing the preconditioning methods and the genetic approach with MSCs.

Point 5: Furthermore, I'm also worried about the novelty of the information in this manuscript– only 16 (13%) out of 128 total references from the last three years (2019-2021), which does not fit well with the dynamics in the field

Response 5: We have tried to review a large proportion of published studies in this topic in order to form a review article resuming the main available data from old and recent studies. New updated references were changed or added to highlight the novelties of the topic (28 out of 143= 19.6% from the past 3 years and 50 out of 143=34.9% from the past 5 years).

Reviewer 3 Report

The topic of the review is interesting and highly topical. The review focuses on different preconditioning methods as well as novel approaches of mesenchymal stem cells therapy. In terms of scientific contribution, the work synthesizes a lot of older and novel information with a clinical aspect. However, I recommend to summarize these described mechanisms in graphical form, which would allow a better understanding for the reader. Similarly, I recommend including synthesis of the results of individual studies in the form of a Table, in particular for the novel approaches. I also recommend a separate section on Future directions of the research, especially from the aspect of human clinical medicine.

Minor comments:

  1. p. 2, line 50: Figure is given in the text, and the Figure is not available in the text

I recommend to check typo errors (eg p. 2, line 59 „futur“, etc ...)

Author Response

Reviewer #3:

Point 1: The topic of the review is interesting and highly topical. The review focuses on different preconditioning methods as well as novel approaches of mesenchymal stem cells therapy. In terms of scientific contribution, the work synthesizes a lot of older and novel information with a clinical aspect.

Response 1: We thank the reviewer for his careful reading of the manuscript, his positive evaluation and his constructive remarks.

Point 2: However, I recommend to summarize these described mechanisms in graphical form, which would allow a better understanding for the reader.

Response 2: A graphical form illustrating the preconditioning methods, novel approaches and the main outcomes was added.

Point 3: Similarly, I recommend including synthesis of the results of individual studies in the form of a Table, in particular for the novel approaches.

Response 3: The wide difference in endpoints and results from the available studies makes difficult to resume it in a single table. Then, we tried to resume findings in the illustrative graphical form that represents the preconditioning methods, novel approaches and the main effects. We also add two tables: one for the preconditioning methods and other for MSCs-genetic modification approach.

Point 4: I also recommend a separate section on Future directions of the research, especially from the aspect of human clinical medicine.

Response 4: A separate section on MSCs perspectives was added.

Point 5: Minor comments:

  1. p. 2, line 50: Figure is given in the text, and the Figure is not available in the text

I recommend to check typo errors (eg p. 2, line 59 „futur“, etc ...)

Response 5: Manuscript was revised by a native English speaker for typos and grammatical errors. Figure was added.

Round 2

Reviewer 1 Report

By adding figures and tables, the manuscript was largely improved. 

- Sentences are still placed without a connection with previous sentences. It is missing a conducting path through the reflection. 

-The English editing was not sufficiently improved: environement (line 53, 70), lines 46-48 are too confusing.

-Still referring to articles too old: for example, lines 63-66 cite an article from 2008. 

-Line 65-66: pro-survival proteins (P65, P50, P105). Which proteins? Aren't P65 and P50 subunits of the nuclear factor-κB?

-Line 80: hypoxia-inducible factor-1α was already changed for HIF-1α in line 64. 

Author Response

Reviewer:

By adding figures and tables, the manuscript was largely improved.

- Sentences are still placed without a connection with previous sentences. It is missing a conducting path through the reflection.

-The English editing was not sufficiently improved: environement (line 53, 70), lines 46-48 are too confusing.

-Still referring to articles too old: for example, lines 63-66 cite an article from 2008.

-Line 65-66: pro-survival proteins (P65, P50, P105). Which proteins? Aren't P65 and P50 subunits of the nuclear factor-κB?

-Line 80: hypoxia-inducible factor-1α was already changed for HIF-1α in line 64.

Response:

We are glad for your positive evaluation, and we are also able to incorporate your new suggestions in the submitted modified version. Herein, the point-by-point response letter:

  • The manuscript was carefully revised for typos and grammatical errors. Connectors were also added.
  • Environment corrected (English editing done). Sentence in line 46-48 was reformulated.
  • We tried to synthetize the maximum proportion of available studies in this topic. We summarize the main result from old and recent studies. In fact, several valuable trials have been conducted in the past and deserve to be cited. We aim through this review article to represent the major published data in this field. However, numerous recent studies have been cited and added in references (28 out 143 (19.6%) between 2019-2022 and 79 out 143 (55%) between 2015-2022).
  • True, the proteins P-50 and P-65 are the sub-units of the nuclear factor-κB. Also, P50 can be generated from P-105. Evidence suggest that these proteins (especially p-50) regulate cells survival. They are considered as pro-survival proteins.
  • After Line 64, HIF-1α was used.

Reviewer 2 Report

The authors have made efforts to revise the manuscript. Now the manuscript has been greatly improved and is generally suitable for publication in Cells. However, the authors should revise the Figure and Tables before it is completely accepted according to Instructions for Authors. Both of the figure and table quality are not good in its present form.

Author Response

Reviewer:

The authors have made efforts to revise the manuscript. Now the manuscript has been greatly improved and is generally suitable for publication in Cells. However, the authors should revise the Figure and Tables before it is completely accepted according to Instructions for Authors. Both of the figure and table quality are not good in its present form.

Response:

  • Figure was improved according to authors instructions (300dpi resolution and format Tiff)
  • Tables were performed according to authors instructions (Created in windows, columns explanatory headings and short title).

Reviewer 3 Report

All comments are sufficiently included in the revised manuscript.

Author Response

We are thankful for your great opinion and previously insightful comments.